# The role of perceived autonomy support and fear of failure: A weekly diary study on work-related rumination

**Elif Manuoglu** *

Department of Psychology, Palacký University Olomouc, Olomouc, Czech Republic

\* manuogluelif@gmail.com

**Data Availability Statement:** All relevant data are within the manuscript and its Supporting Information files.

**Funding:** This publication was made possible thanks to targeted funding provided by the Czech

## Abstract

Grounded in self-determination theory, the present study examined the weekly fluctuations in different forms of work-related rumination depending on perceived autonomy support and fear of failure at the workplace. Work-related rumination has three dimensions, affective rumination (negative emotions or affect), problem-solving pondering (thinking over the actions to handle the problems), and psychological detachment (mentally distancing oneself from work during nonwork time). In total, 111 employees ($M_{age}$ = 34.88, $SD$ = 10.43) from various occupations were followed over the course of three weeks via weekly measurements, resulting in 333 matched observations. Multilevel random coefficient modeling showed that on the weeks when employees reported higher levels of perceived autonomy support from the leader, they engaged in affective rumination and problem-solving pondering less. However, weekly fluctuations in psychological detachment from work was not associated with perceived autonomy support. Moreover, on the weeks when employees experienced high fear of failure, they reported less psychological detachment from work during nonwork time. Lastly, within-and and between-person fear of failure moderated the negative link between perceived autonomy support and affective rumination. Findings showed that perceived autonomy support is a protective factor for employees high in both state and trait fear of failure in decreasing affective rumination. Directions for future research and implication for practice were discussed.

## Introduction

It has been well-documented that workplace stressors give rise to rumination (e.g., [1–4]). Previous research mainly examined the link between work-related rumination and various psychological and physiological well-being outcomes (e.g., [5–7]) and it has been demonstrated that work-related rumination is a detrimental process to employees' well-being. To be able to meet the work demands without rumination and the following negative outcomes, employees may need some nutrients, such as supportive work environments during work and psychological detachment from work during nonwork time. Grounded in self-determination theory [8, 9], the present study focuses on the antecedents of work-related rumination with a weekly

Ministry of Education, Youth, and Sports for specific research, granted in 2022 to Palacký University Olomouc (IGA_FF_2022_018).

**Competing interests:** The author have declared that no competing interests exist.

diary design. Since the type of rumination rather than rumination per se is critical due to their differential relationships with well-being [5], three concepts of work-related rumination were employed in the current study to examine the weekly changes in each type based on perceived autonomy support, an essential nutrient for employees in the workplace, and fear of failure, a form of performance anxiety. Both autonomy support and fear of failure play a significant role in boosting or dampening well-being, which necessitates exploring their role on the weekly changes in work-related rumination to be able to provide effective means of decreasing work-related rumination.

This study aims to enhance our understanding of factors that are increasing and decreasing work-related rumination. It is proposed that perceived autonomy support from supervisors would be negatively associated with work-related rumination. Moreover, fear of failure would be positively linked to work-related rumination. Lastly, trait fear of failure would moderate the negative link between perceived autonomy support and affective rumination.

The current study makes several contributions to work-related rumination literature. First, it aims to extend the current understanding of the role of perceived autonomy support and fear of failure as a workplace stressor on work-related rumination during non-work time. Second, the present study aims to show whether perceived autonomy support can have a protective role for employees with high fear of failure. This means that if there is such an effect, some practical solutions can be offered to decrease affective rumination based on the framework of self-determination theory. Last, as the current study has a weekly diary design, it is possible to demonstrate whether changes in perceived autonomy support and fear of failure predict changes in work-related rumination.

## Defining work-related rumination

Rumination is a broad term that refers to a variety of persistent thinking processes and is defined as conscious and perseverative thinking about a common instrumental theme in the absence of environmental demands necessitating this thinking [10]. Rumination includes cognitive and emotional processes that refer to an active thinking process about a stressor and the feelings and thoughts arising from that stressor accompanied by its future implications. Ruminators think that perseverative thinking about daily life issues is useful [11].

Work-related rumination is a form of rumination that extends work demands outside of work. It is defined as having persistent thoughts directed to unresolved issues about work in the absence of work demands [12]. The effects of work-related issues can last long even when the situation causing them is not available after work [13]. In the current study, three forms of work-related rumination were included in accordance with the distinction proposed by Cropley and Zijlstra (2011) [12]. These are affective rumination, problem-solving pondering, and lack of psychological detachment.

Affective rumination is characterized by experiencing intrusive, pervasive, and recurrent thoughts about problems, their causes, and potential consequences. Attention is directed toward the negative feelings related to problems [14]. This type of thinking is often related to recurrent thinking but not solving problems [15]. Consequently, affective rumination is considered a negative process since it brings about negative emotional states. Accordingly, it was associated with lower well-being [2, 6].

In contrast, problem-solving pondering is characterized by solving problems related to work. The recurrent thinking process is understood in terms of rehearsing and thinking over the necessary actions to handle the problem at hand. Accordingly, problem-solving pondering lacks the negative affective quality, is unemotional, and includes prolonged thinking about the solutions to work-related problems [12]. Thus, the outcomes of problem-solving pondering

can be positive if a solution can be reached at the end of the process. Moreover, it is goal-directed because even if the process does not result in a practical solution, pondering may lead to developing a specific plan or strategy for dealing with the problem.

Lastly, psychological detachment is defined as abstaining from work-related activities and mentally distancing oneself from work during nonwork time. Psychological detachment or mentally switching off work is a requirement for the recovery process during the off-work time [16]. Detachment is characterized by not being preoccupied with job-related tasks, such as receiving phone calls at home or actively participating in job-related activities. It also includes being mentally involved in other content, such as leisure activities [17]. Psychological detachment is seen as one way to stop or reduce ruminative thinking [10]. Previous studies showed that the lack of psychological detachment is related to lower well-being [16, 17].

The previous research showed that although these three concepts are correlated [17, 18], they seem to tap different aspects of work-related rumination [19]. Moreover, Querstret and Cropley (2012) [5] suggested that the type of rumination rather than rumination per se can be crucial due to their differential associations with recovery. Therefore, three concepts of work-related rumination are used in the current study to examine the weekly changes in each type during three weeks and the specific mechanism underlying the relationship between perceived autonomy support and work-related rumination. Although there is a wealth of evidence on the detrimental effects of work-related rumination on well-being, little is known about how autonomy support from leaders and fear of failure at work contribute to work-related rumination. For example, a recent meta-analysis provided indirect evidence and demonstrated that perceived autonomy support from leaders is negatively correlated with employee psychological distress and work-specific stress [20]. The current study examines the link between perceived autonomy support at work and work-related rumination from the framework of self-determination theory.

## Linking perceived autonomy support and work-related rumination from the self-determination theory perspective

Self-Determination Theory (SDT; [8, 9]) is a macro theory of human motivation that focuses on the degree to which behaviors of individuals are autonomously motivated or controlled. It has been applied across various life domains, including research on work organizations [21–23]. The theory claims that social environments (e.g., workplaces) can support or hinder individuals' efforts toward self-determination [24].

Research showed that autonomy support has been associated with a number of positive outcomes, such as more intrinsic motivation and interest, less pressure/tension, higher self-esteem, more creativity and cognitive flexibility, and a more positive emotional tone [25]. An autonomy-supportive context refers to interpersonal contexts where individuals are seen as independent and capable of self-determination. Thus, autonomy support is also different from permissiveness (lack of structure) and neglect (lack of involvement) [26].

Contextual factors facilitating employee autonomy and self-determination are also the key focus in the organizational domain. Job autonomy is often considered a contextual antecedent of self-determination [27]. Past research showed that the interpersonal orientation that managers have toward their employees could be a critical antecedent of job autonomy [28]. In that study, it was demonstrated that managers' motivational style, which can range from highly autonomous to highly controlling, was associated with the perception, affect, and satisfaction of the employees. Research in work organizations has tended to focus on either the perspectives of employees or the perspectives of leaders. The perspectives of employees were focused on in the current study.

In the work context, autonomy support from leaders refers to a cluster of supervisory behaviors that promote a supportive and understanding environment within the leader-employee relationship [29]. In that respect, perceived autonomy support is the perception of employees that supervisors try to acknowledge their perspectives, offer choices, provide informative feedback rather than controlling and negative feedback about their performances, provide rationales for requested behaviors, and promote self-initiation [30]. Perceived autonomy support from leaders is an essential nutrient for employees to feel secure and free to perform their duties at work in a manner that fits their personal values and preferences [20, 30]. Employees feel as the active agent and regulators of their behaviors while performing a work activity if their supervisors are supportive [31]. Past research showed that autonomy support in the work context had been associated with less anxiety [32], less psychological distress, role ambiguity, role overload [33], and more subjective well-being [22].

In contrast, a controlling leader gives controlling feedback rather than informative feedback, uses controlling language while communicating, and treats poor performance as a problem to be solved rather than a focus to evaluate [28]. Employees having controlling leaders are pressured to feel, act, and think in particular ways. They also feel that their feelings and needs are ignored [27]. Furthermore, when employees deviate from the demands of the leader, they face corrective or some punitive actions aimed at restoring the behavior back to its planned trajectory [28]. Consequently, having high-quality relationships with supervisors almost becomes impossible, and employees may tend to ruminate about these negative feelings and experiences after work. This is not surprising as affective rumination is characterized by focusing on negative aspects of work, such as failures [34]. Moreover, it has been shown that individuals ruminate about not only negative emotional states but also factors contributing to those emotional states [35]. For example, an employee who has a debate with her/his supervisor at work may continue to think about that encounter and the supervisor himself/herself after work. Thus, not only a negative experience at work but also a controlling supervisor can lead to affective rumination.

For affective rumination, it is likely that employees who have a controlling leader are unable to disengage from work-related issues after a tough week. For example, receiving controlling and negative feedback or receiving strict instructions from the leader on a task can direct the attention of employees toward the negative feelings related to that feedback or instructions during non-work time. To the best of our knowledge, no study to date has directly tested the relation of perceived autonomy support to work-related rumination from a self-determination theory perspective. Yet, in a recent meta-analysis, it was shown that negative or toxic work events are significantly associated with work-related rumination [36]. As the previous examples suggested, a controlling leader can be accounted for some negativity at work. Based on this information, it was hypothesized that perceived autonomy support is negatively associated with affective rumination. More specifically, on the weeks when employees report more perceived autonomy support from their leaders, they also report less affective rumination (H1).

It has been shown that psychological detachment or mentally switching off from work is associated with job stressors negatively and is crucial for preventing rumination [7]. The prolonged thinking about job stressors makes psychological detachment from work difficult [16]. As mentioned previously, a controlling supervisor can also be a critical stressor through its effects on employees' perceptions, leading to a lack of psychological detachment. A supportive supervisor, on the other hand, can enhance the psychological detachment from work during nonwork time. For example, a supportive supervisor can enhance the autonomy of employees by acknowledging employees' way of completing a particular task, which decreases the pressure on the employees. As a result, employees do not need further thinking about the task in nonwork time. Based on this, it is proposed that perceived autonomy support is positively

associated with psychological detachment. More specifically, on the weeks when employees report more perceived autonomy support from their leaders, they also report more psychological detachment from work (H2).

Lastly, problem-solving pondering can result in positive outcomes if it results in a solution. Thus, the results of problem-solving pondering depend on the situation and it is less certain. Besides, employees may not have that chance, or even if they have, the occasions can be rare in the course of the study. Consequently, a specific hypothesis regarding the link between perceived autonomy support and problem-solving pondering is not formulated. The analyses for problem-solving pondering are exploratory.

Given the lack of research in this area, there is a need to expand our current knowledge of the mechanisms between perceived autonomy support and work-related rumination. Therefore, the moderating role of fear of failure is discussed in the following sections. The present study aimed to uncover the situational conditions under which employees benefit from perceived autonomy support.

## Fear of failure and work-related rumination

Fear of failure is operationalized as a form of performance anxiety. It is an anxiety-induced state in which individuals experience intense fear and self-doubts regarding the consequences of a failure. Employees are preoccupied with the fear of being viewed as incompetent in the eyes of others [37, 38]. Although fear of failure has been primarily studied in academic and research contexts (e.g., [39–41]), job-related fear of failure has been studied especially in the last years (e.g., [42]).

Although there is no research examining the direct link between fear of failure and work-related rumination, some prior research showed that they could be related. In a past experimental study, it was shown that fear of failure was one of the most worried and ruminated topics among participants who had been identified as high-ruminators [43]. Hjeltnes et al. (2015) [44] also showed that fear of academic failure is a concept that participants could ruminate about. According to the conceptualization of Conroy (2001) [45], expecting feelings of shame and embarrassment after a failure, beliefs about losing future opportunities, devaluating one's social value and work performance, and upsetting others are common indicators among individuals who associate with threat appraisal and fear. To give a concrete example, when an employee is confronted with a challenging situation, fear of failure and preoccupations with self-doubt would be higher. This situation can lead to avoidance of future failure or pursuit associated with the following shame and embarrassment, which in turn may give rise to decreased psychological detachment from work. In other words, it is possible that an employee recurrently thinks about the possible reactions of others and the negative feelings (i.e., fear) associated with this problem even after work. That is, intrusive thoughts following the experience of a negative event can continue during the off-job time, which prevents recovery after work [7] and can also increase affective rumination. As remembered, controlling supervisors impose their own ideas and solutions to problems without paying attention to the perspectives of employees, provide negative feedback, and use controlling language with phrases like "must" and "should". Consequently, it is likely that employees would feel intense pressure and fear of not meeting the requirements of a task when faced with an achievement situation or pursuit of a challenging activity at the workplace due to the reactions of their supervisors or others in the workplace.

Although fear of failure is conceptualized as a trait-like construct in previous research (e.g., [42], it can also show fluctuations in a weekly manner depending on the demands of the work in a given week. Based on these arguments, it is expected that weekly changes in fear of failure

will predict weekly changes in psychological detachment and affective rumination (H3). More specifically, on the weeks when employees report more fear of failure, they also report less psychological detachment from work and more affective rumination. A specific hypothesis was not formulated regarding the link between fear of failure and pondering due to a lack of evidence in the literature concerning this link. Thus, the analysis was exploratory.

Furthermore, the trait fear of failure may moderate the negative link between weekly perceived autonomy support and affective rumination. That is, the individualized relationship with the leader can decrease work-related rumination more for people who have higher levels of self-doubts about the consequences of a failure. Based on these arguments, it was reasoned that perceived autonomy support could discourage affective rumination for people with high trait fear of failure more effectively. Thus, it was hypothesized that the trait fear of failure would moderate the negative link between perceived autonomy support and affective rumination (H4).

## The present study

Drawing upon self-determination theory, the present study aimed to examine the weekly changes of three types of work-related rumination based on perceived autonomy support and fear of failure. Rumination is a multifaceted construct and the antecedents of different dimensions can vary necessitating the exploration of the antecedents of the dimensions separately [46]. Building on previous work, this research fills the gap in the literature by providing a comprehensive examination of different types of work-related rumination from the perspective of self-determination theory, a macro theory of well-being. The following hypotheses were formulated:

H1: Affective rumination would be lower on the weeks when employees report higher perceived autonomy support.

H2: Psychological detachment would be higher on the weeks when employees report higher levels of perceived autonomy support.

H3: Psychological detachment would be lower and affective rumination would be higher on the weeks when employees report higher levels of fear of failure.

H4: Fear of failure would moderate the negative association between perceived autonomy support and affective rumination. Specifically, it was proposed that perceived autonomy support could decrease affective rumination of people with higher levels of trait fear of failure. The moderation model is displayed below (Fig 1).

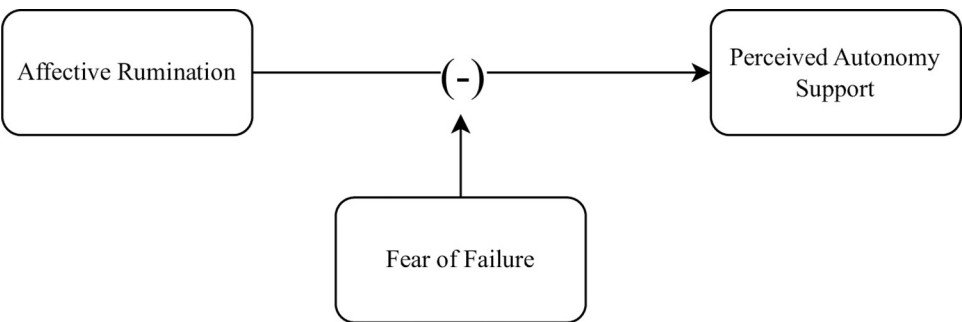

**Fig 1. Proposed moderation model.**

In the current study, all variables are conceptualized as both state-like (fluctuations over weeks) and trait-like (stable characteristics) concepts. In this way, specific situational factors triggering work-related rumination can be determined over and above the stable individual differences. Moreover, fear of failure depends on many factors, such as particular conditions of the week. For example, it may not be possible to experience fear of failure on a daily basis due to a number of reasons in the workplace, such as task distribution. Moreover, since emotions, behaviors, or performances of individuals tend to be dynamic across different occasions depending on the circumstances [47], a weekly diary design can capture these fluctuations. Therefore, employing a weekly design is the ideal method in the present study.

## Method

### Participants, procedure, and materials

Respondents were employees from different occupational sectors (e.g., education, health, tourism) but mainly from the education sector both in Turkey (38%) and Czechia (16%). The next highest percentage of the sector is 2% with marketing and the other sectors have around 1% ratio. To be eligible as a participant, employees should work full-time in white-collar jobs. Initially, the study was advertised on various platforms (e.g., social media, billboards) by research assistants and the personal networks of the author. Three hundred and eight participants (147 from Czechia) completed the baseline survey. They were contacted for the weekly surveys via email. Almost one-third of the participants completed all surveys. In total, there were 111 participants (333 matched observations) (70 from Turkey, 41 from Czechia) who completed all measures (baseline + three consecutive weeks). More than half of the participants were female (68.5%; CZ = 69% female, TR = 68% female) and from Turkey (62.2%). The average age was 34.88 years (SD = 10.43; $CZ_{age}$ = 39.27, SD = 12.48; $TR_{age}$ = 33.87, SD = 8.46). The average tenure with the organization was 75.58 months (*SD* = 80.73; CZ = 83.64, SD = 92; TR = 70.67, SD = 73). The majority of the participants in both samples have a university degree (CZ = 45.2%, TR = 43.5%).

The Ethics Review Board of the Middle East Technical University approved the study. Full completion of the study involved three phases: receiving online information about weekly survey procedures, completion of a baseline questionnaire, and three consecutive weeks of records. Firstly, employees from various organizations were contacted and given information about the study, including information about the procedures and confidentiality. The baseline questionnaire was also sent in these emails. The informed consent form was provided to participants in baseline questionnaires. Upon agreement, participants received weekly surveys following three consecutive weeks. All participants completed the baseline questionnaire between Friday evening at 6 p.m. and Sunday morning. Participants were requested to use their email as their identifier code in the weekly surveys to be able to match their responses. Instead of asking them to create a unique code, they were asked to write their emails because participants tend to forget even these unique codes or change the ordering of the code and confuse the matching procedure. In weekly surveys, participants were requested to consider their previous week and complete the questionnaires accordingly. Reminder emails were sent to participants on Monday mornings. The data were collected in six waves. The same procedures were applied to all waves by trained research assistants.

**Work-related rumination.**   Three forms of work-related rumination were measured with the Work-Related Rumination Scale developed by Cropley et al., (2012) [13]. The Turkish version of the work-related rumination scale, translated by Akyuz and Sulak (2019) [48], was administered to Turkish participants. For Czech participants, the scale was translated into Czech via a standard translate-back-translate approach. A faculty member from the

department of psychology who is fluent both in English and Czech language translated the items first, and then two research assistants, who are also fluent in English and Czech language, back-translated the items. After checking the items in terms of clarity and meaning, the last versions of the items were decided together with the author and the research assistants. Affective rumination was measured with five items. A sample item: "Do you become tense when you think about work-related issues during your free time?". Prob"em-solving pondering was measured with five items. An example item from the scale: "I find thi"king about work during my free time helps me to be creative." Lastly, psychological detachment was also measured with five items. A sample item from the scale: "I make myself switch off from work as soon as I leave." The response scale ranged from 1 (*very seldom or never*) to 7 (*very often or always*). Participants were asked to respond to the items with respect to their off-job time. Cronbach's α is .88 for affective rumination and .83 for pondering and detachment in this study. The same items were used in the weekly surveys.

*The work climate questionnaire*. The Work Climate Questionnaire [30] was used to measure employees' perceptions regarding managerial autonomy support. The Turkish version of the scale which had been translated and used by the author of this study for another unpublished project was used for Turkish participants. For the Czech participants, the back-translation procedure was applied to translate the survey items into the Czech language. It is a 15-item scale, and responses are made on a 7-point scale ranging from 1 (*strongly disagree*) to 7 (*strongly agree*). A sample item from the scale: "I feel that my manager provides me choices and options." Higher average scores represent a higher level of perceived autonomy. Cronbach's α of the scale is .96 in this study. The shortened version of the scale with six items was used in weekly surveys not to cause exhaustion in participants with the long version. Items were selected based on their representativeness of the state level of autonomy support from the leader. These items were 1, 2, 4, 5, 8, and 12.

**Fear of failure at work.** Fear of failure was measured with Houston and Kelly's (1987) [49] fear of failure scale modified by Elliot and Church (2003) [50]. This scale was translated into Turkish and the Czech language by the same translation-back-translation procedure employed previously. It is a 7-item measure and each item is rated on a 1 (*strongly disagree*) to 7 (*strongly agree*) scale. A sample item from the scale: "When I fail at a task, I am even more certain that I lack the ability to perform the task.". Higher scores indicate a higher level of fear of failure. Cronbach's α is .84 in this study. In the weekly version, the same seven items were used.

**Demographic information.** Demographic information form were presented in the baseline questionnaire. Gender (female = 0, male = 1), age, country (TR = -1, CZ = 1), sector, and tenure with the current job were used as control variables.

The following sentence was added to the beginning of weekly surveys: "During the previous week. . ."

## Data analytic strategy

The data of the present study entail a multilevel structure because weekly measurements are nested within individuals. A series of multilevel random coefficient models were used to analyze the data in HLM 8.00. All Level 1 variables were group-mean centered (i.e., an individual's score of a predictor variable represents an individual's deviation from her/his own mean score across three weeks) and Level 2 variables were grand-mean centered (i.e., an individual's score of a predictor variable represents individual's deviation from the grand mean of the entire sample) (see [51] for a discussion of centering in multilevel models). Demographic variables (age, sex, and tenure with the organization) and baseline level of outcome variables (trait level

**Table 1. Descriptive statistics and intercorrelations of the study (baseline) variables.**

|  | M | SD | 1 | 2 | 3 | 4 | 5 | 6 | 7 | 8 |
|---|---|---|---|---|---|---|---|---|---|---|---|
| 1. Age | 35.88 | 10.43 | - |  |  |  |  |  |  |  |
| 2. Gender |  |  | -.18 | - |  |  |  |  |  |  |
| 3. Tenure | 75.58 | 80.73 | .51** | -.19 | - |  |  |  |  |  |
| 4. PAS | 5.14 | 1.38 | .03 | -.06 | .01 | - |  |  |  |  |
| 5. Fear | 3.49 | 1.18 | -.30** | .02 | -.06 | -.02 | - |  |  |  |
| 6. Affective | 3.87 | 1.44 | -.06 | .08 | -.01 | -.21* | .30** | - |  |  |
| 7. Pondering | 4.06 | 1.28 | .15 | .07 | .16 | .09 | .04 | .63** | - |  |
| 8. Detachment | 3.74 | 1.40 | -.03 | .12 | -.18 | -.04 | -.29** | -.58** | -.49** | - |

*Note*. PAS. = Perceived Autonomy Support; Fear. = Fear of Failure; Affective = Affective Rumination

*p < 0.05

**p < 0.01

or stable individual differences) were controlled in all analyses to show that the weekly effects of perceived autonomy support and fear of failure on the weekly work-related rumination are over and above one's trait level work-related rumination and demographics. Before the analyses, data were screened for outliers and there were no outliers.

## Results

### Testing hypothesized models

Descriptive statistics and correlations for study variables are displayed in Table 1 (for baseline variables) and Table 2 (for weekly variables). As in previous research ([52, 53] correlations for weekly variables are calculated for each week and the highest and lowest correlations are presented in Table 2.

### Within-person associations between work-related rumination and perceived autonomy support and fear of failure

The initial analyses tested null models for each dependent variable (no predictors at Level 1 or Level 2) and estimated within- and between-person variances. The intraclass correlation (ICC) was calculated based on these variances for each dependent variable [55]. An example model is presented below. Analyses showed that ICC is .70 for weekly affective rumination, reflecting between-person variance. Within-person variation accounts for the remaining 30% of the total variance (1-ICC). Thus, affective rumination does not only vary between individuals but also differs within individuals on a weekly basis. ICC for pondering is .62 and .64 for detachment.

**Table 2. Descriptive statistics and intercorrelations of the weekly variables.**

|  | M | SD | 1 | 2 | 3 | 4 | 5 |
|---|---|---|---|---|---|---|---|
| 1. PAS | 4.89/5.16 | 1.63/1.43 | - |  |  |  |  |
| 2. Fear | 2.88/3.15 | 1.31/1.31 | .02/.03 | - |  |  |  |
| 3. Affective | 3.62/3.70 | 1.50/1.50 | -.11/-.20* | .28**/.46** | - |  |  |
| 4. Pondering | 3.92/3.96 | 1.32/1.24 | .18/.25** | .07/.22** | .58**/.63** | - |  |
| 5. Detachment | 4.00/4.10 | 1.41/1.56 | .02/.13 | -.22*/-.28** | -.61**/-.69** | -.46**/-.56** | - |

*Note*. *p < 0.05

**p < 0.01

Therefore, multilevel modeling is appropriate to analyze the data because there is a considerable amount of variance that is left to be explained by within-person fluctuations. The following equation was used to test the null models.

Level 1 Model $AffectiveRumination_{ti} = \pi_{0i} + e_{ti}$

Level 2 Model $\pi_{0i} = \beta_{00} + r_{0i}$

Briefly, the equations estimating weekly work-related rumination as a function of weekly perceived autonomy support and fear of failure were constructed at the within-person level (Level 1 Model) and at the between-person level (Level 2 Model). Such analyses are equal to estimating a regression coefficient for each person and then analyzing these coefficients [54]. A sample equation from these analyses is presented below. In this example, the significance of mean coefficients representing an association between outcomes (i.e., work-related rumination) and predictor (e.g., perceived autonomy support) was tested by examining whether the $\beta_{10}$ coefficient was significantly different from 0. These coefficients are interpreted similarly to coefficients from OLS regression analyses [54].

Level 1 Model $AffectiveRumination_{ti} = \pi_{0i} + \pi_{1i}*(PAS_{ti}) + e_{ti}$

Level 2 Model $\pi_{0i} = \beta_{00} + \beta_{01}*(AGE_i) + \beta_{02}*(SEX_i) + \beta_{03}*(TENURE_i) + \beta_{04}*(AffeR_i) + r_{0i}$

$\pi_{1i} = \beta_{10} + \beta_{11}*(AGE_i) + \beta_{12}*(SEX_i) + \beta_{13}*(TENURE_i) + \beta_{14}*(AffeR_i) + r_{1i}$

First, perceived autonomy support was included at Level 1 to investigate the within-person associations with work-related rumination. Analyses were performed for affective rumination, pondering, and detachment separately. Guidelines suggested by Nezlek (2008) [55] were followed and a forward-stepping procedure was employed to build the models. For example, perceived autonomy support and fear of failure were tested in different models. If two of them were significant in predicting the same outcome, they were entered together into the same model.

Analyses demonstrated that weekly perceived autonomy support is negatively associated with weekly affective rumination ($\beta_{10} = -.39$, $t = -3.75$, $p < .001$) (H1) and pondering ($\beta_{10} = -.34$, $t = 2.44$, $p < .01$). In other words, on the weeks when participants reported more autonomy support from their leader, they also report less affective rumination and problem-solving pondering. However, there was no significant association between psychological detachment and perceived autonomy support ($\beta_{10} = -.22$, $t = -1.55$, $p = .125$), although the relationship was in the expected direction (H2). The same analyses were performed for fear of failure as a Level 1 predictor and work-related rumination as the outcome. Analyses revealed that weekly fear of failure is associated with weekly affective rumination ($\beta_{10} = .31$, $t = 2.01$, $p < .05$), and detachment ($\beta_{10} = -.33$, $t = -2.50$, $p < .05$) (H3), but not with pondering ($\beta_{10} = .06$, $t = .36$, $p = .71$). The coefficients of these analyses are demonstrated in Tables 3 and 4. Thus, the first and third hypotheses were supported. Moreover, variability across the slopes of the link between affective rumination and perceived autonomy support was displayed in Fig 2, as an example. As it is seen, relationship slopes vary considerably across employees.

**Table 3. Relationships between perceived autonomy support and work-related rumination controlling for demographics.**

| Rumination | Age | Sex | Tenure | PAS |
|---|---|---|---|---|
| Affective | .01 | .20 | .01* | -.39*** |
| Pondering | -.01 | .26 | .01 | -.34* |
| Detachment | .01 | .17 | -.01 | -.22 |

*Note.* Coefficients accompanied by

* were significantly different from 0 at $p < .05$, and

*** were significantly different from 0 at $p < .001$. PAS = Perceived Autonomy Support

**Table 4. Relationships between fear of failure and work-related rumination controlling for demographics.**

| Rumination | Age | Sex | Tenure | Fear |
|---|---|---|---|---|
| Affective | .01 | -.05 | -.01 | .31* |
| Pondering | -.01 | -.10 | -.01 | .06 |
| Detachment | -.01 | -.02 | .01 | -.33* |

*Note.* Coefficients accompanied by

\* were significantly different from 0 at $p < .05$. Fear = Fear of Failure

Since weekly fear of failure was a significant predictor of weekly affective rumination, it was included in the same model with perceived autonomy support in predicting affective rumination. Both fear of failure ($\beta_{20} = .36$, $t = 2.536$, $p < .05$) and perceived autonomy support ($\beta_{10} = -.41$, $t = -3.96$, $p < .001$) were significant in predicting affective rumination in the same model. Although it was not hypothesized, state fear of failure (within-person level) was used as a moderator on the link between perceived autonomy support and affective rumination. The same level interaction term between weekly perceived autonomy support and fear of failure was created in SPSS and entered into the model in HLM, while perceived autonomy support and fear of failure were also in the model. Analysis showed that this interaction is significant ($\beta_{30} = -.43$, $t = -2.77$, $p < .01$). That is, the weekly negative impact of perceived autonomy support on affective rumination was greater for individuals who reported higher levels of weekly fear of failure.

Lastly, the link between the trait fear of failure and work-related rumination was tested. Trait fear of failure (between-person level) predicted weekly psychological detachment ($\beta_{01} = -.33$, $t = -3.20$, $p < .01$), affective rumination ($\beta_{01} = .42$, $t = 4.91$, $p < .001$), and pondering ($\beta_{01} = .16$, $t = 1.96$, $p = .054$), which showed that individuals with higher trait fear of failure are

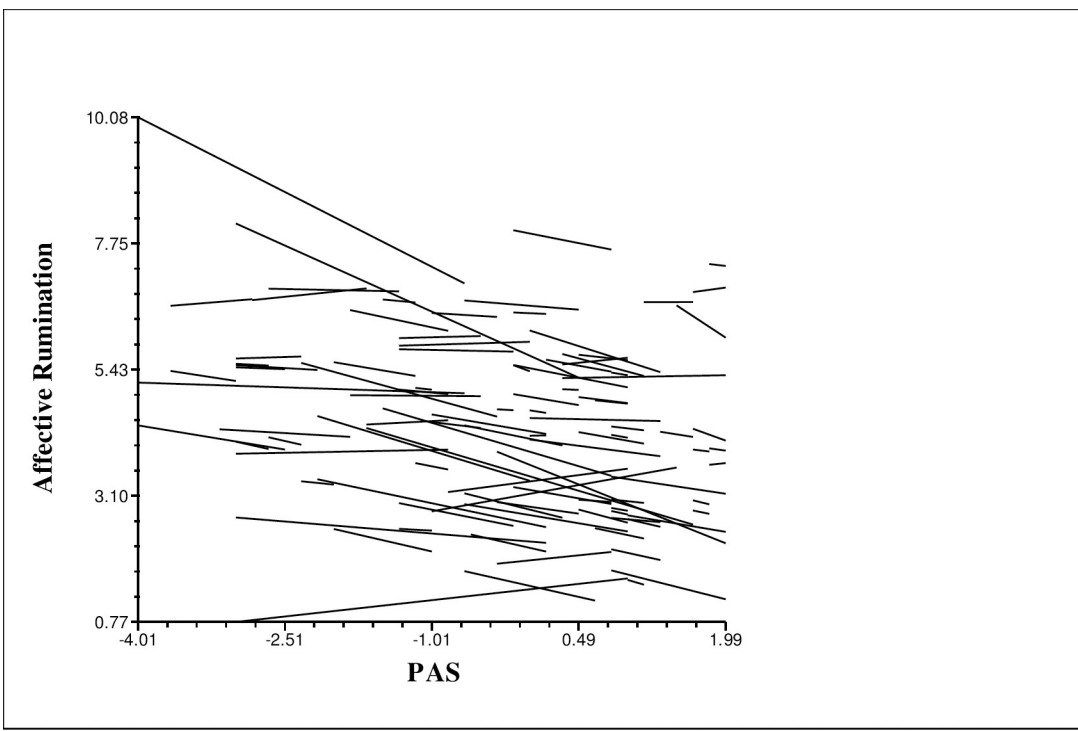

**Fig 2. Relationship slopes of the link between affective rumination and perceived autonomy support.**

more vulnerable to affective rumination and pondering, and experience less psychological detachment.

### Trait-level measure of fear of failure and its moderating role in the link between weekly perceived autonomy support and work-related rumination

Whether within-person relationships between weekly measures affective rumination and perceived autonomy support varied as a function of fear of failure, the stable individual difference variable, was tested by analyzing within-person slopes (H4). Tenure with the organization was used as a control in these analyses as it can be related to fear of failure. The equation is presented below:

Level 1 Model (*AffectiveR*) $\gamma_{ij} = \pi_{0j} + \pi_{1j}*(PAS_{ij}) + e_{ij}$

Level 2 Model $\pi_{0j} = \beta_{00} + \beta_{01}*(Tenure_j) + \beta_{02}*(FearofFailure_j) + r_{0j}$

$\pi_{1j} = \beta_{10} + \beta_{11}*(Tenure_j) + \beta_{12}*(FearofFailure_j) + r_{1j}$

In multilevel modeling, such analyses are referred to as cross-level interactions or slopes as outcomes analyses in which relationships at one level of analysis (i.e., state level or Level 1) vary as a function of a variable at another level of analysis (i.e., trait level or Level 2) [56]. $B_{02}$ represents the relationship between fear of failure and the intercept for each person (i.e., mean weekly affective rumination), and $\beta_{12}$ represents the relationship between fear of failure and perceived autonomy support slope for each individual. Thus, the $\beta_{12}$ coefficient was examined to see if the moderation is significant. Analyses showed that the negative link between weekly affective rumination and weekly perceived autonomy support is strongest for individuals with the highest fear of failure (i.e., +1 SD above the mean) ($\beta_{12} = -.12$, $t = -2.06$, $p < .05$) (see Fig 3). The nature of this cross-level interaction is consistent with the nature of the same-level interaction between perceived autonomy support and fear of failure. For exploratory purposes, the moderating role of fear of failure on the link between perceived autonomy support and other forms of work-related rumination (pondering and detachment) was also tested but there was no significant effect.

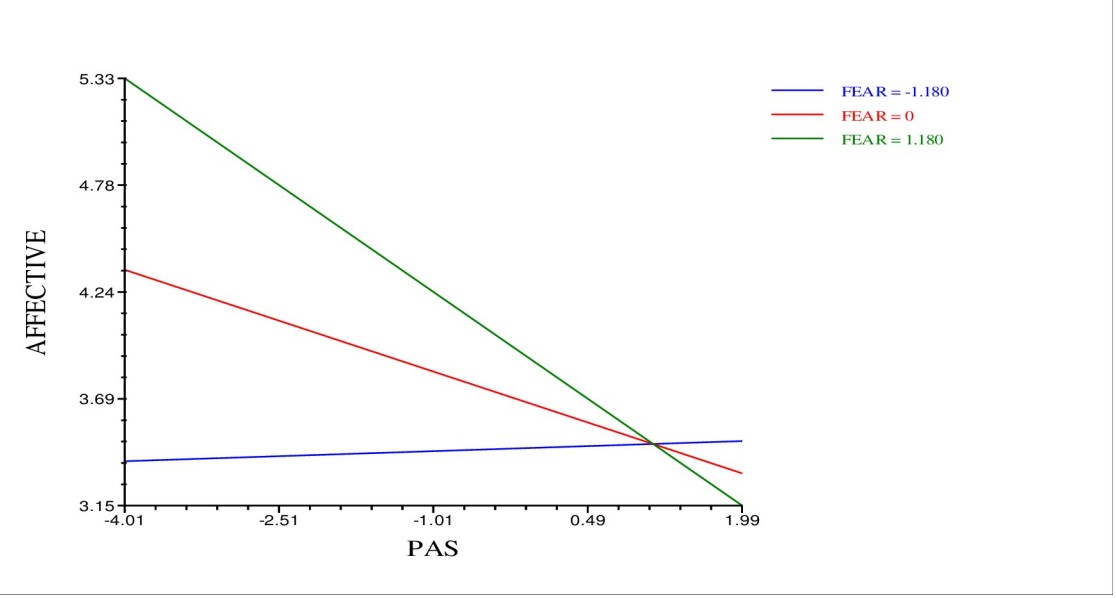

**Fig 3. Moderating role of state fear of failure on the link between perceived autonomy support and affective rumination.** The moderating effect is graphed for three levels of fear of failure: +1 SD above the mean, mean, and -1 SD below the mean.

## Exploratory analysis

**Country-level differences in within-person relationships between work-related rumination and predictors.** The differences between Turkish and Czech samples in terms of within-person associations between perceived autonomy support/fear of failure and work-related rumination were tested by including a contrast-coded variable (TR = 1 and CZ = -1) in Level 2. Analyses showed that there was not any significant difference between the two samples regarding within-person relationships between perceived autonomy support/fear of failure and work-related rumination. An example model is displayed below:

Level 1 Model $AffectiveRumination_{ti} = \pi_{0i} + \pi_{1i}*(PAS_{ti}) + e_{ti}$

Level 2 Model $\pi_{0i} = \beta_{00} + \beta_{01}*(COUNTRY_i) + r_{0i}$

$\pi_{1i} = \beta_{10} + \beta_{11}*(COUNTRY_i) + r_{1i}$

## Discussion

The present study aimed to examine the weekly fluctuations in work-related rumination based on perceived autonomy support and fear of failure. It was expected that there would be a negative association between perceived autonomy support and affective rumination and a positive association between psychological detachment. Moreover, there would be a negative link between fear of failure and psychological detachment. Lastly, the trait fear of failure would moderate the negative relationship between perceived autonomy support and affective rumination. Although the link between pondering and perceived autonomy support was exploratory, most of the findings were in line with the expectations. The findings also revealed that perceived autonomy support has a protective role for employees high in trait and state fear of failure in terms of decreasing affective rumination.

## Weekly fluctuations in work-related rumination: The role of perceived autonomy support

As expected, weekly changes in affective rumination were predicted by perceived autonomy support (H1). On the weeks when employees perceived that their leaders were supporting their autonomy, they engaged in less affective rumination. The findings suggest that when employees feel that their leaders respect their perspectives and feelings, their way of completing a task, convey confidence in their ability to perform a task, and care for them as a person [30], they also engage in less affective rumination. Although there is no previous research examining this link directly, there are a number of research demonstrating the negative link between managerial autonomy support and negative job-related outcomes such as absenteeism and turnover intentions [56], exhaustion and role conflict [57] and distress [20]. New evidence provided in the current study shows that autonomy support can also mitigate affective rumination, a negative state after work. Thus, autonomy support from the leader is a necessary nutrient for employees to decrease affective rumination.

Contrary to expectation, perceived autonomy support did not predict weekly changes in psychological detachment (H2). In other words, perceived autonomy support did not improve the psychological detachment experience of employees. This finding suggests that employees need different nutrients for psychological detachment from work. Weekly changes in problem-solving pondering, on the other hand, were predicted by perceived autonomy support. More specifically, on the weeks when employees perceived high autonomy support, they were less likely to engage in problem-solving pondering. Previous research mainly conceptualized problem-solving pondering as a positive process (e.g., [12, 18]) and demonstrated associations negatively with burnout [58], counterproductive behavior [59], and chronic work-related

fatigue ([5]; and positively with creativity at work [59, 60]. However, there are also studies showing the link between problem-solving pondering and negative job-related outcomes, such as work pressure [6], boundary-crossing behavior [2], and emotional exhaustion [60]. The present study demonstrated that some features of an autonomous leader (e.g., providing choices, accepting and caring about employees, and having trust toward employees) decreased problem-solving pondering. As mentioned previously, the outcomes of problem-solving pondering can be beneficial in case practical solutions are reached at the end of the process [18]. However, the findings of the present study suggest that perceived autonomy support may mitigate the need to engage in problem-solving-pondering. It should be noted that this analysis was exploratory and no hypothesis was formulated about this link. Given the link between problem-solving pondering and perceived autonomy support at the workplace has scarcely been examined, and the previous findings are mixed, this finding should be interpreted cautiously. More research in this area is needed.

## The role of fear of failure

As stated in the third hypothesis, weekly changes in fear of failure predicted weekly changes in psychological detachment and affective rumination. That is, on the weeks when employees felt more fear of failure, they also reported less psychological detachment from work and more affective rumination in their nonwork time. Findings suggest that when employees have recurrent thoughts about the possible reactions of others and the negative feelings linked to a particular problem or task, they continue to have these intrusive thoughts even after work. When preoccupied with the fear of failure, employees' recovery from work decreases and affective rumination increases which, in turn, is detrimental to well-being (e.g., [7, 61, 62].

Findings also suggest that there is a spillover effect of fear of failure in the workplace that is transmitted to nonwork time and decreases psychological detachment and increases affective rumination. The spillover effect is defined as the transfer and interference of reactions experienced in work to a nonwork domain [63]. Since data were completed only three weeks, it is unpractical to perform lagged analyses to see whether the previous week's fear of failure at work predicts the next week's psychological detachment in this study. However, it is apparent that the relationship of fear of failure with psychological detachment occurs within a close timeframe, such as in the same week. In other words, employees transmitted their fear of failure from work to their nonwork time as evidence of spillover relationships. The present study showed that fear of failure could also change depending on the demands of the job, even though it is conceptualized as a trait-like construct (e.g., [42]). Thus, the present study showed that fear of failure varies not only between individuals but also within individuals on a weekly basis.

## The moderating role of trait and state fear of failure in the link between perceived autonomy support and affective rumination

This study also explored the moderating role of fear of failure on the link between different forms of work-related rumination and perceived autonomy support. As expected, it was found that greater perceived autonomy support from leaders was more strongly related to lower affective rumination in individuals high in both trait and state fear of failure (H4). More specifically, weekly perceived autonomy support decreased weekly affective rumination most for people who reported higher levels of weekly fear of failure. For the trait fear of failure, the same pattern was observed. On the weeks when individuals reported higher levels of perceived autonomy support, they also reported lower levels of affective rumination, but most for individuals with higher levels of trait fear of failure. The moderating role of fear of failure was

presented both at the within- and between-person levels in this study. The apparent explanation is that perceived autonomy support protects against maladaptive recurrent thoughts about work, to which individuals who are higher in both trait and state fear of failure are prone. The findings claim that perceived autonomy support, which fosters adaptability [64], autonomous work motivation [20], self-initiation [30], and affective commitment to the organization [65], can enhance and create individualized perceptions of meaningful relationships between employees and leaders, and pleasant work environments. The findings of the present study showed that the abovementioned characteristics of autonomy support effectively decrease the affective rumination of employees with higher levels of fear of failure.

Thus, the combination of perceived autonomy support and fear of failure can be a critical factor in decreasing affective rumination. To discourage affective rumination, supporting autonomy is especially imperative for those employees who have higher levels of fear of failure. Conversely, a weaker relationship between perceived autonomy support and affective rumination was observed for employees with lower levels of fear of failure, suggesting that employees with lower levels of fear of failure are less sensitive to the experience of autonomy support. These employees may be insulated from this process regardless of their trait level of perceived autonomy support. Different mechanisms may link perceived autonomy support to positive work-related outcomes, such as job/career satisfaction and lower intention to quit. In these types of relationships, it is possible that the effects might be stronger for employees with lower levels of fear of failure. This can be an avenue for future research.

Indeed, the utility of perceived autonomy support in the workplace has been well-documented so far (e.g., [20, 28, 29], to date. The present study corroborated the previous findings and provided additional evidence about the protecting role of perceived autonomy support for individuals with higher levels of state and trait fear of failure by adopting the self-determination theory perspective. The study extended the current understanding of which employees benefit most from their experience of autonomy support in the workplace in terms of discarding their engagement in affective rumination. Finally, as I proposed in the current study, the different dimensions of work-related rumination indeed were associated with fear of failure differently, but not with perceived autonomy support. These findings supported the importance of examining the dimensions of work-related rumination in relation to different antecedents of well-being separately for more comprehensive and precise conclusions.

### Limitations and directions for future research

Despite its strengths, this study is not without its limitations that should be taken into account while evaluating the findings. First, the direction of causality cannot be discerned since performing lagged analyses was not practical because of the number of weeks. Inverse causality between the variables could potentially be considered and longitudinal studies with more weeks can address this question. Second, although employees were recruited from a wide variety of occupations, they are all white-collar workers and may differ from blue-collar workers in some aspects. It can be interesting to examine these differences. Thus, future research can recruit blue-collar workers to test the links between work-related rumination and perceived autonomy support and fear of failure. Third, the associations examined in this study have not been previously tested. Thus, more studies are needed to validate the findings of this study. Lastly, only one source of autonomy support was measured in the present study. Different sources of autonomy support may have relative importance to employees. For example, autonomy support from other sources (e.g., colleagues) was found as an essential nutrient for employees in terms of job-related outcomes [65, 66]. Examining the role of autonomy support from different sources on work-related rumination can be another avenue for future research.

In addition to autonomy support, future research can examine the role of other positive nutrients in the work environment that can be a protective factor against the fear of failure and diminish affective rumination. Appreciation, for example, can be a good candidate since it is associated with positive work outcomes, such as increased motivation [67], positive psychological functioning [68], and well-being [69]. Therefore, appreciation from different sources can be examined in relation to fear of failure and work-related rumination in future studies. Lastly, only leader autonomy support was measured as a work-related positive nutrient but there can be a nonwork-related nutrient for employees to decrease affective rumination and increase detachment. For example, engaging in leisure activities can be helpful for employees since previous research has documented that leisure pursuits are effective means of increasing psychological detachment [70]. Therefore, leisure crafting can be another protective factor for employees with high fear of failure to increase detachment. This can be another avenue for future research to examine.

## Implications for practice and conclusion

In the present study, the importance of perceived autonomy support in decreasing affective rumination and protective role for employees high in trait and state fear of failure was demonstrated for the first time. In addition to its relation with positive individual work outcomes (e.g., [20]), perceived autonomy support is a crucial nutrient for decreasing negative work-related outcomes. The present findings have implications for designing intervention programs for a healthy work environment. The findings suggest that leaders can be trained in terms of providing autonomy support on a regular basis (e.g., once a year) to ensure its permanence by integrating such an intervention into the organizational program. Although other sources of autonomy support were not examined in this study, it would also be beneficial to train employees in providing autonomy support to each other in the workplace, which can be as effective as leader autonomy support in terms of positive results, as previous research suggested [65]. However, participating in training activities should be voluntary, as pointed out by Querstret et al. (2016) [15]. If it is enforced, the very first step in autonomy support is violated. Finally, the exploratory analysis showed that although Turkey and Czechia have different backgrounds in terms of work-related practices [71], within-person associations between perceived autonomy support/fear of failure and work-related rumination did not differ between them. In other words, within-person associations between the variables were country-independent in that study. However, the proposed associations were tested for the first time in this study, which should be taken into consideration while evaluating the findings.

The present study also underscores the importance of conceptualizing fear of failure separately as a trait and state fear of failure. Since state fear of failure showed fluctuations on a weekly basis in this study, it can be decreased through some effective steps, such as autonomy support, as evidenced in this study. In this way, employees can engage in less affective rumination, detach themselves from work, and experience an effective recovery after work.

## Supporting information

**S1 File.**
(SAV)

**S2 File.**
(SAV)

## Acknowledgments

The author thanks Dr. Ersin Kara and two research assistants, Vendula Macalova and Richard Smilnak, for their support during data collection.

## Author Contributions

**Conceptualization:** Elif Manuoglu.

**Data curation:** Elif Manuoglu.

**Formal analysis:** Elif Manuoglu.

**Funding acquisition:** Elif Manuoglu.

**Investigation:** Elif Manuoglu.

**Methodology:** Elif Manuoglu.

**Project administration:** Elif Manuoglu.

**Software:** Elif Manuoglu.

**Writing – original draft:** Elif Manuoglu.

**Writing – review & editing:** Elif Manuoglu.

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
