## [Decision Letter · Decision Letter 0]

26 Jul 2023

PONE-D-22-27725The role of perceived autonomy support and fear of failure: A weekly diary study on work-related ruminationPLOS ONE

Dear Dr. Manuoğlu,

Thank you for submitting your manuscript to PLOS ONE. After careful consideration, we feel that it has merit but does not fully meet PLOS ONE’s publication criteria as it currently stands. Therefore, we invite you to submit a revised version of the manuscript that addresses the points raised during the review process. The reviewers' comments are appended at the bottom of this email.  Please submit your revised manuscript by Sep 09 2023 11:59PM. If you will need more time than this to complete your revisions, please reply to this message or contact the journal office at plosone@plos.org. Please include the following items when submitting your revised manuscript:A rebuttal letter that responds to each point raised by the academic editor and reviewer(s). You should upload this letter as a separate file labeled 'Response to Reviewers'.A marked-up copy of your manuscript that highlights changes made to the original version. You should upload this as a separate file labeled 'Revised Manuscript with Track Changes'.An unmarked version of your revised paper without tracked changes. You should upload this as a separate file labeled 'Manuscript'.

We look forward to receiving your revised manuscript.

Kind regards,

Sharjeel Saleem

Academic Editor

PLOS ONE

Journal Requirements:

Reviewers' comments:

Reviewer's Responses to Questions

**Comments to the Author**

1. Is the manuscript technically sound, and do the data support the conclusions?

Reviewer #1: Partly

Reviewer #2: Yes

2. Has the statistical analysis been performed appropriately and rigorously? 

Reviewer #1: Yes

Reviewer #2: Yes

3. Have the authors made all data underlying the findings in their manuscript fully available?

Reviewer #1: Yes

Reviewer #2: Yes

4. Is the manuscript presented in an intelligible fashion and written in standard English?

Reviewer #1: Yes

Reviewer #2: Yes

5. Review Comments to the Author

Reviewer #1: I am grateful to you for sending in your work. The following are some remarks that I have regarding this paper:

1. This article's most significant challenges in attempting to justify sampling from two distinct nations. It is of the utmost need to provide a rationale for why the study, which has a smaller sampling size from two separate countries, should be accorded any significance.

2. It is necessary for the author to provide a clear definition of remuneration psychology. The entire amount of pay an employee receives in exchange for the work they do for their company, and the theory that explains how this works, could be understood by the general public if certain conditions are met. In most cases, this takes the form of monetary benefits, which may also be referred to as a wage or salary and can vary from nation to nation, depending on that country's economic realities. However, some supplementary benefits are becoming more and more popular. Remuneration mechanisms can be set, but it is possible to compare one state's demographics to those of another state. How does this make sense?

3. We were unable to determine the ranks, genders, cadres, years of experience, or industries that the interviewees belonged to. As a result, drawing conclusions about the nature of responses based on this information is challenging. Author may incorporate demographic profile of respondents from Turkey and Czechia

Reviewer #2: This study explores the influence of perceived autonomy support and fear of failure in the workplace on weekly fluctuations in different forms of work-related rumination. The manuscript is commendably well-written and structured. The statistical analyses demonstrate a high level of technical proficiency and are sufficiently described.

Nevertheless, there is room for improvement, and I offer the following suggestions to enhance the manuscript:

1. Introduction Section: The significance of the study needs to be clearly stated. Also, identify the research gap that this study aims to fill with at least one reference. How the current study builds upon prior research and addresses unanswered questions? For example, it may be right that there is lack of research in the area (heading: Perceived autonomy support and work-related rumination, last para), but lack of research is not a strong claim in itself. It is suggested to include how researchers have encouraged scholars to explore the effects of the relationships under study and then cite those researchers who have mentioned that these relationships need to studied.

2. While the study's foundation on the self-determination theory is appropriate and aligns well with the framework, it is advisable to expand the theoretical perspective beyond a single heading, such as "self-determination theory." To strengthen the research, consider incorporating the theoretical perspective more explicitly during hypothesis development and discussion. Additionally, acknowledge the study's theoretical contribution in the discussion and highlight it as a significant aspect of the overall research findings.

3. I would suggest to provide separate and well-defined research questions or hypotheses to precisely outline the study's objectives. While research hypotheses are mentioned, they are currently merged within the text. To enhance readability and facilitate the reader's understanding, it is recommended to state the research hypotheses explicitly and with more clarity. This will help the reader grasp the primary research objectives more easily.

4. While reviewing the manuscript, I noticed only one graphical diagram. To enhance clarity and facilitate readers' understanding, it is advisable to add more visual diagrams, particularly a model diagram that illustrates the study's framework.

5. Under the heading "The Present Study” (second para), it would be beneficial to include at least one reference to support the claim that a weekly diary study is the ideal method for this research. Providing a reference will reinforce the suitability and effectiveness of this study design.

6. Under the heading “Within-person associations between work-related rumination and perceived autonomy support and fear of failure” (2nd last para, last line), the statement seems to describe a moderating effect but the wording is slightly confusing. I would suggest to rephrase the sentence for better clarity and understanding.

7. Under the heading "Participants, Procedure, and Material," (2nd line), it would be beneficial to cite a relevant study that specifies the eligibility criteria used in this research.

8. Under the heading "Method," in the subheading "Work-related rumination," on the 3rd line, consider using an alternative word to "developed" for clarity (for example, ‘translated’ or ‘adapted’).

9. Under the heading "The moderating role of trait and state fear of failure in the link between perceived autonomy support and affective rumination," in the first paragraph, second-to-last line, there is a typo - 'abovementioned'.

10. Lastly, I would suggest to update the references. While there is one reference from 2022, adding more up-to-date references will make the study more current and show a better understanding of the latest research in the field.

It has been a pleasure reviewing this manuscript and I look forward to seeing this published soon. Good luck!

---

## [Author Response · Author response to Decision Letter 0]

9 Aug 2023

Reviewer #1: I am grateful to you for sending in your work. The following are some remarks that I have regarding this paper:

1. This article's most significant challenges in attempting to justify sampling from two distinct nations. It is of the utmost need to provide a rationale for why the study, which has a smaller sampling size from two separate countries, should be accorded any significance.

Thank you for raising this crucial point and creating a space for me to explain it. Although it was not my focal aim, I wanted to show that if there are differences between countries in terms of within-person associations between perceived autonomy support/fear of failure and work-related rumination, then I would have proof that work-related rumination also fluctuates depending on the country. I performed exploratory analyses to test this in the last set of analyses. Due to the challenge of collecting data from employee samples, we could reach 111 participants (333 matched observations) in total (70 from Turkey). It seems that maybe I should have tried to reach more employees, especially from CZ but due to project time limitations and that employees were about to enter the summer holiday season, I had to stop the data collection. 

Although these cultures have different characteristics in terms of work-related practices, I found a country-independent relationship between the variables when I used the country as a moderator in Level 2 equations. In other words, there was not any significant difference between the two samples from Czechia and Turkey concerning within-person relationships between perceived autonomy support/fear of failure and work-related rumination. I should have elaborated on that issue with a few sentences in the discussion section. I added a brief part under the “Implications for practice and conclusion” section as follows: Finally, the exploratory analysis showed that although Turkey and Czechia have different backgrounds in terms of work-related practices [74], within-person associations between perceived autonomy support/fear of failure and work-related rumination did not differ between Turkey and Czechia. In other words, within-person associations between the variables were country-independent.”

2. It is necessary for the author to provide a clear definition of remuneration psychology. The entire amount of pay an employee receives in exchange for the work they do for their company, and the theory that explains how this works, could be understood by the general public if certain conditions are met. In most cases, this takes the form of monetary benefits, which may also be referred to as a wage or salary and can vary from nation to nation, depending on that country's economic realities. However, some supplementary benefits are becoming more and more popular. Remuneration mechanisms can be set, but it is possible to compare one state's demographics to those of another state. How does this make sense?

Thank you for pointing out that important issue. The majority of the respondents from both countries are teachers (education sector) sharing similar working conditions. And it was reported that work-related rumination is common among teachers since they have to spend some time after work on work-related tasks (e.g., Türktorun et al., 2020; Wu et al., 2023). Thus, I added the following sentences to the method section: “Respondents were employees from different occupational sectors (e.g., education, health, tourism) but mainly from the education sector both in Turkey (38%) and Czechia (16%). The next highest percentage of the sector is 2% with marketing and the other sectors have around 1% ratio.” Moreover, according to OECD 2022 data (https://data.oecd.org/teachers/teachers-salaries.htm), annual teacher salaries with 15 years of experience are close in the Czech Republic and Turkey (28534 $ vs. 35365 $). Thus, I do not think that pay grades could explain significant variance in work-related rumination in that study and I did not want to emphasize this issue. Furthermore, salary information is a Level 2 variable and doesn't fluctuate (like gender, age, and tenure, which were controlled in the analyses). However, I aim to see the fluctuations in work-related rumination by controlling for the demographics. Thus, although involving salary information makes sense in organizational research, it wouldn't be a crucial variable in the present study which stresses the role of perceived autonomy support and fear of failure. I would have controlled it in the analyses if I measured it. However, it was a shortcoming that I did not ask in the survey.

Turktorun, Y. Z., Weiher, G. M., & Horz, H. (2020). Psychological detachment and work-related rumination in teachers: a systematic review. Educational Research Review, 31, 100354.

Wu, Q., Cao, H., & Du, H. (2023). Work Stress, Work-Related Rumination, and Depressive Symptoms in University Teachers: Buffering Effect of Self-Compassion. Psychology Research and Behavior Management, 1557-1569.

3. We were unable to determine the ranks, genders, cadres, years of experience, or industries that the interviewees belonged to. As a result, drawing conclusions about the nature of responses based on this information is challenging. Author may incorporate demographic profile of respondents from Turkey and Czechia

Thank you so much for pointing out that issue. I added a brief part in the “Participants” section and reported the characteristics of the two samples from Turkey and Czechia separately. 

In general, I would like to thank the first reviewer for her/his insightful feedback that led me to think deeply about the study in general and revise it accordingly. I really appreciated that the feedback concerned the whole paper in general, not only some specific parts. I will benefit from her/his feedback in my future research designs as well. 

Reviewer #2: This study explores the influence of perceived autonomy support and fear of failure in the workplace on weekly fluctuations in different forms of work-related rumination. The manuscript is commendably well-written and structured. The statistical analyses demonstrate a high level of technical proficiency and are sufficiently described.

Nevertheless, there is room for improvement, and I offer the following suggestions to enhance the manuscript:

1. Introduction Section: The significance of the study needs to be clearly stated. Also, identify the research gap that this study aims to fill with at least one reference. How the current study builds upon prior research and addresses unanswered questions? For example, it may be right that there is lack of research in the area (heading: Perceived autonomy support and work-related rumination, last para), but lack of research is not a strong claim in itself. It is suggested to include how researchers have encouraged scholars to explore the effects of the relationships under study and then cite those researchers who have mentioned that these relationships need to studied.

Thank you for raising this crucial issue. The following brief part was added under the first section, at the end of the first paragraph to state the aim of the the study clearly: “Since the type of rumination rather than rumination per se is critical due to their differential relationships with well-being [5], three concepts of work-related rumination were employed in the current study to examine the weekly changes in each type based on perceived autonomy support, an essential nutrient for employees in the workplace, and fear of failure, a form of performance anxiety. Both autonomy support and fear of failure play a significant role in boosting or dampening well-being, which necessitates exploring their role on the weekly changes in work-related rumination to be able to provide effective means of decreasing work-related rumination.”

Moreover, the following sentence was added to the “Present Study” section to state how this research fills the gap in the literature: “Rumination is a multifaceted construct and the antecedents of different dimensions can vary necessitating the exploration of the antecedents of the dimensions separately [60]. Building on previous work, this research fills the gap in the literature by providing a comprehensive examination of different types of work-related rumination from the perspective of self-determination theory, a macro theory of well-being..”

2. While the study's foundation on the self-determination theory is appropriate and aligns well with the framework, it is advisable to expand the theoretical perspective beyond a single heading, such as "self-determination theory." To strengthen the research, consider incorporating the theoretical perspective more explicitly during hypothesis development and discussion. 

Thank you for this feedback. This section titled “self-determination theory” was merged with the following section and the new title is as follows: “Linking perceived autonomy support and work-related rumination from the self-determination theory perspective”. Moreover, the previous “self-determination theory” section has been shortened to make a good fit with the following section and for a smooth transition.

I also incorporated the self-determination theory perspective explicitly in the different parts of the discussion section. For the hypothesis development, I did not do it because it would be a repetition of the previous sections and I did not want to have so much repetition. 

Additionally, acknowledge the study's theoretical contribution in the discussion and highlight it as a significant aspect of the overall research findings.

The following part was added to the discussion section to emphasize the theoretical contribution of the study as an overall finding: “Finally, as I proposed in the current study, the different dimensions of work-related rumination indeed were associated with fear of failure differently, but not with perceived autonomy support. These findings supported the importance of examining the dimensions of work-related rumination in relation to different antecedents of well-being separately for more comprehensive and precise conclusions.”

3. I would suggest to provide separate and well-defined research questions or hypotheses to precisely outline the study's objectives. While research hypotheses are mentioned, they are currently merged within the text. To enhance readability and facilitate the reader's understanding, it is recommended to state the research hypotheses explicitly and with more clarity. This will help the reader grasp the primary research objectives more easily.

Following this feedback, I created a hypothesis part under the “Present study” section. I wrote the hypotheses one by one to make them more visible. I also changed the wording of some hypotheses to make them more understandable. This section is as follows: “The following hypotheses were formulated: 

H1: Affective rumination would be lower on the weeks when employees report higher perceived autonomy support. 

H2: Psychological detachment would be higher on the weeks when employees report higher levels of perceived autonomy support. 

H3: Psychological detachment would be lower on the weeks when employees report higher levels of fear of failure. 

H4: Fear of failure would moderate the negative association between perceived autonomy support and work-related rumination (affective rumination and psychological detachment). Specifically, it was proposed that perceived autonomy support could decrease affective rumination of people with higher levels of trait fear of failure.”

4. While reviewing the manuscript, I noticed only one graphical diagram. To enhance clarity and facilitate readers' understanding, it is advisable to add more visual diagrams, particularly a model diagram that illustrates the study's framework.

I added two new figures (Figure 1 and Figure 2) to enhance clarity and facilitate readers’ understanding. Figure is about the basic model of the present study and Figure concerns the relationships slopes of the link between affective rumination and perceived autonomy support.

5. Under the heading "The Present Study” (second para), it would be beneficial to include at least one reference to support the claim that a weekly diary study is the ideal method for this research. Providing a reference will reinforce the suitability and effectiveness of this study design.

The necessary adjustment was done and a reference was added in that paragraph. 

6. Under the heading “Within-person associations between work-related rumination and perceived autonomy support and fear of failure” (2nd last para, last line), the statement seems to describe a moderating effect but the wording is slightly confusing. I would suggest to rephrase the sentence for better clarity and understanding.

For the interest of brevity, I deleted this sentence because it was already like a repetition of the previous sentence.

7. Under the heading "Participants, Procedure, and Material," (2nd line), it would be beneficial to cite a relevant study that specifies the eligibility criteria used in this research.

Thank you for the comment. Although the eligibility criteria is specific to this study, it is also common in organizational research to recruit participants with full-time jobs. I am interested in employees having full-time jobs to be able to measure work-related rumination over the long run. Part-time jobs can have other issues that I cannot control in the course of the study. For example, employees with part-time jobs can have more than one job and perceived autonomy support would be problematic to measure. For the white-collar issue, I already added a limitation concerning why I only reached out to employees with white-collar jobs. Thus, eligibility criteria is common among other research and no need to give a citation. 

8. Under the heading "Method," in the subheading "Work-related rumination," on the 3rd line, consider using an alternative word to "developed" for clarity (for example, ‘translated’ or ‘adapted’).

Thank you for the correction. I replaced the word with “translated” as you suggested. 

9. Under the heading "The moderating role of trait and state fear of failure in the link between perceived autonomy support and affective rumination," in the first paragraph, second-to-last line, there is a typo - 'abovementioned'.

Thank you for the comment. It is not a typo, there is a word “abovementioned”, that refers to something that was mentioned earlier. I wrote it on purpose.

10. Lastly, I would suggest to update the references. While there is one reference from 2022, adding more up-to-date references will make the study more current and show a better understanding of the latest research in the field.

Thank you for the comment. Although I couldn’t find a study relevant to the aims of the current study, I added one study (Wu & Zhou, 2023) testing the link between workplace stressors and rumination. 

Thank you for the all comments that improved this paper further! It has been a pleasure to follow these feedback and work on them. 

It has been a pleasure reviewing this manuscript and I look forward to seeing this published soon. Good luck!

---

## [Decision Letter · Decision Letter 1]

29 Aug 2023

The role of perceived autonomy support and fear of failure: A weekly diary study on work-related rumination

PONE-D-22-27725R1

Dear Dr. Manuoğlu,

We’re pleased to inform you that your manuscript has been judged scientifically suitable for publication and will be formally accepted for publication once it meets all outstanding technical requirements.

Kind regards,

Sharjeel Saleem

Academic Editor

PLOS ONE

Additional Editor Comments (optional):

Reviewers' comments:

Reviewer's Responses to Questions

**Comments to the Author**

1. If the authors have adequately addressed your comments raised in a previous round of review and you feel that this manuscript is now acceptable for publication, you may indicate that here to bypass the “Comments to the Author” section, enter your conflict of interest statement in the “Confidential to Editor” section, and submit your "Accept" recommendation.

Reviewer #1: (No Response)

Reviewer #2: All comments have been addressed

2. Is the manuscript technically sound, and do the data support the conclusions?

Reviewer #1: (No Response)

Reviewer #2: Yes

3. Has the statistical analysis been performed appropriately and rigorously? 

Reviewer #1: Yes

Reviewer #2: Yes

4. Have the authors made all data underlying the findings in their manuscript fully available?

Reviewer #1: Yes

Reviewer #2: Yes

5. Is the manuscript presented in an intelligible fashion and written in standard English?

Reviewer #1: Yes

Reviewer #2: Yes

6. Review Comments to the Author

Reviewer #1: (No Response)

Reviewer #2: (No Response)

7. PLOS authors have the option to publish the peer review history of their article (what does this mean?). If published, this will include your full peer review and any attached files.

Reviewer #1: No

Reviewer #2: No

---

## [Editor Report · Acceptance letter]

26 Sep 2023

PONE-D-22-27725R1 

The role of perceived autonomy support and fear of failure: A weekly diary study on work-related rumination 

Dear Dr. Manuoğlu:

I'm pleased to inform you that your manuscript has been deemed suitable for publication in PLOS ONE. Congratulations! Your manuscript is now with our production department. 

Kind regards, 

on behalf of

Dr. Sharjeel Saleem 

Academic Editor

PLOS ONE